# OpenReview forum: "The Hidden Dimensions of LLM Alignment: A Multi-Dimensional Analysis of Orthogonal Safety Directions"
_ICML.cc/2025/Conference — ICML 2025 poster_

### Official Review · Reviewer_6Hnk · 2025-02-14

**Overall Recommendation:** 4

**Summary:**

**Post-rebuttal edit: the authors have provided detailed and convincing responses to my concerns during the discussion phase, meaning I'm happy to increase my score from 3 to 4. I believe this paper deserves to be at ICML.**

---

This work performs a multi-dimensional analysis of the shift in representations $\mathcal{T}(\textbf{x})$ induced by safety fine-tuning in language models. It does so by approximating this shift as an affine transformation $\textbf{W}\textbf{x}+\textbf{b}$ then analysing the mathematical properties (e.g. principal components, effective rank) of the matrix $\textbf{W}$. Attempts are then made to relate the principal components to safety-relevant semantic concepts (e.g. refusal), including via the layerwise relevance propagation (LRP) method for token attribution. It is found that the first principal component predicts refusal (albeit somewhat less reliably than a dedicated probe vector for this behaviour) while subsequent ones are thought to represent secondary features such as role-playing. Interventional attack analyses are then used to demonstrate the causal effect of these directions on model behaviour.

**Claims And Evidence:**

The claims of this paper can be broadly divided into:
1. descriptive claims about the geometry of representation shifts during fine-tuning;
2. causal claims about the impact of specific shift directions on model behaviour.

The former are largely predicated on the accuracy of the linear shift approximation, but this is not empirically verified (see "Methods And Evaluation Criteria"). The latter seem to be quite well supported by the intervention experiments, although I do have some concerns and questions about these (see "Experimental Designs Or Analyses").

**Essential References Not Discussed:**

None.

**Experimental Designs Or Analyses:**

Overall, the experimental design is well-structured and easy to follow. I have a few comments about the presentation:
- The LRP results seem to mainly focus on a single handcrafted example, so your interpretations may be overfitted. It would be good to see an aggregated analysis across many examples.
- Figure 4 shows that ablating the L14-C6 direction selectively impairs the model's response to PAIR attacks, but lacks baselines against ablating other directions. These baselines would be helpful to show that L14-C6 is especially influential.
- Figure 6 shows that the attacks change the projection onto the dominant component. In the text you say that this "leads to a lower refusal rate", but this refusal rate is not actually plotted in Figure 6. I feel this should be included.

**Methods And Evaluation Criteria:**

The method begins by learning the linear approximation $\textbf{W}\textbf{x}+\textbf{b} \approx \mathcal{T}(\textbf{x})$ then performs an analysis on this approximation. In itself, I find this to be quite an interesting idea, but I do have two important concerns:

- Nowhere in the paper can I *how accurate this linear approximation is*. You make a few references to the linear representation hypothesis, but from my understanding, this hypothesis does not imply anything about how representations can shift during fine-tuning. How accurate are the approximations in these experiments?

- The entire analysis is performed on the matrix $\textbf{W}$, with no consideration of the bias vector $\textbf{b}$. Surely if we are to understand the directions in which representations shift, this bias term could be quite important? I would appreciate it if you could address this in your response. If there is some reason why it is reasonable to ignore the bias term, I would be very happy to hear about it and update my opinion of the method accordingly.

**Other Comments Or Suggestions:**

- There are typos (e.g. "experiemental") and grammatical errors (e.g. "Imagine you conducting") in the handcrafted example in Table 1.
- In Section 5.2, you say that Figure 3 shows that "the model's harmfulness prediction accuracy plateaus after layer 15", but this is a plot of *refusal* prediction accuracy. This comment therefore seems misleading.
- The baseline methods for jailbreak prompts are not discussed anywhere in the main body of the paper. I suggest you add a sentence briefly introducing these, and pointing to Appendix C.1 for more details.

**Other Strengths And Weaknesses:**

None; all mentioned in other sections.

**Questions For Authors:**

1. Can you respond to my concern about the accuracy of the linear approximation? ("Methods And Evaluation Criteria" section)
2. Can you respond to my concern about the omision of $\textbf{b}$ from the analysis? ("Methods And Evaluation Criteria" section)
3. When discussing Figure 2, you say how the effective ranks of $1$ in early layers suggests "that safety training neither introduces nor strengthens new directions -- this aligns with the mid-early safety layer hypothesis". I don't quite follow this argument. Could you put it in different terms, perhaps with more explicit reference to the underlying geometry?
4. Why is the actual refusal rate not plotted in Figure 6? Currently the reader has to take your word for it that the attacks actually modify the model behaviour in the expected way.

**Relation To Broader Scientific Literature:**

This paper falls within the area of mechanistic interpretability for language models. By focusing more specifically on interpreting *changes* in model internals as a result of fine-tuning, it attempts to tackle an under-explored and safety-relevant problem.

**Theoretical Claims:**

This work is primarily empirical in nature. One theoretical aspect is Theorem 3.2, and I somewhat doubt its utility. You start Section 3.1 by saying that you're interested in "whether the components in the residual space contain interpretable features", but then (after giving the theorem) admit that it "does not guarantee human-interpretable features". In that case, I really don't see what benefit it provides, other than give an air of theoretical rigor to what is ultimately an empirical exercise.

---

> ### Author Rebuttal · Authors · 2025-03-31
>
> We thank the reviewer for the insightful comments. We are encouraged that the reviewer acknowledges our novelty and well-designed experiments. We will carefully address your concerns below:
>
> > **Q1:** How accurate are the approximations in these experiments?
>
> You raised a good point. We did not initially include the accuracy of the residual space map because our framework does not strictly require the activation transformation $T(X)$ to be entirely linear. However, we agree with you that it would be problematic if the map only poorly approximated $T(X)$ with low accuracy.
>
> We provide the Mean Square Error evaluation for approximations in the following table. We found the MSE between finetuned activations and the approximation is negligible compared to the Mean Square Norm of activations. This suggests that **in our safety training case, the learned affine map accurately captures $T(X)$.**
>
> **[[ View table of MSE]](https://shorturl.at/e6ac4)**
>
> We will discuss and report this result in the revised paper.
>
> > **Q2:** The entire analysis is performed on the matrix W, with no consideration of the bias vector b.
>
> You raised a very good question. **We excluded bias vector b from this analysis because it does not affect extracted feature directions**. We study feature directions in this paper, which are directions in activation space activated by certain context features. Since **b** represents the component of the affine map independent of the activation $X$, it acts as a constant offset regardless of the input. Hence, the direction represented by **b** is not a feature direction as defined and is not addressed by our methods, although we agree it might carry useful semantic meaning.
>
> Alternatively, performing SVD and selecting the right singular vectors finds the *directions with the largest variance* in the residual space. As **b** is the global offset, it does not affect variance.
>
> > **Q3:** Figure 4 shows that ablating the L14-C6 direction selectively impairs the model's response to PAIR attacks, but lacks baselines against ablating other directions.
>
> You raised a valid point. In the revised paper, **we have updated Figure 4 to include results for L14-C4 and L14-C5 to show the selective impact of L14-C6**. Please see the figure and caption below.
>
> **[[View updated Figure 4]](https://cdn-uploads.huggingface.co/production/uploads/62cd3a3691d27e60db0698b0/ZzTJiZ5HC52b8cqa9E1g3.png)**
>
> >**Q4:** Effective ranks of 1 in early layers suggests "that safety training neither introduces nor strengthens new directions...". I don't quite follow this argument.
>
> When the effective rank is 1, the variance in the activation transformation is mainly captured by a single direction, indicating that activations are changing along a 1-d line. Compared to the much higher ranks (around 100) observed in mid-to-late layers, we conclude that no substantial learning of new features happens in the early layers.
>
> > **Q5:** Refusal rate is not actually plotted in Figure 6. I feel this should be included.
>
> We initially omitted the refusal rate from Figure 6 due to (1) figure design considerations and (2) this metric primarily reflects output behavior rather than showing internal representation changes. However, we agree that including this data will strengthen the presentation of our results.  We provide the refusal rate after intervention on the test set in the table below. We can see non-dominance suppression significantly reduces the refusal rate for jailbroken samples, while having a small impact on harmful and benign samples.
>
> **[[View table of refusal rate in Figure 6]](https://cdn-uploads.huggingface.co/production/uploads/62cd3a3691d27e60db0698b0/sBYs5rTPEAspyW1k10cjI.png)**
>
> We will discuss and provide these results in Section 6.
>
>
> > **Q6:** It would be good to see an aggregated analysis across many examples for LRP results.
>
> In the paper, we used a handcrafted example to showcase PLRP results mainly for visualization purposes, as we found it difficult to understand relevant tokens without concrete context. . **The following table demonstrates that the top relevant tokens remain interpretable even when aggregated across the test set.**
>
> **[[View table of aggregated evaluation]](https://cdn-uploads.huggingface.co/production/uploads/62cd3a3691d27e60db0698b0/1DO2yA6CUI_lJU6zez-Y6.png)**
>
> We will provide this aggregated evaluation in the revision
>
> > **Q7:** Theorem 3.2 Utility
>
> You raised a really good point. We agree that our primary claims are supported by empirical evidence. **Our theorem mainly provides *a motivation for later analysis***, suggesting that the residual space can be analyzed similarly to feature directions. Our experimental results support the theorem and provide deeper insights beyond the theorem statement. In the revised version, we will explicitly discuss the purpose of our theorem.
>
> > **Q8:** Other Comments Or Suggestions
>
> Thank you for pointing these out! We will address these points in the updated version.

---

> > ### Comment · Reviewer_6Hnk · 2025-04-01
> >
> > Thanks for your response. I've replied only where I feel I have more to add (all on the topic of the bias term).
> >
> > **Q2:** This is helpful, but I'm still not quite satisfied here. I don't follow you claim that *"the direction represented by $\textbf{b}$ is not a feature direction"*. Would it not be meaningful to measure the extent to which $\textbf{b}$ aligned with (for example) a "harmlessness" direction, in which case we could say that fine-tuning tends to shift all activations to be more harmless?
> >
> > **Q4:** You say that an effective rank of $1$ indicates that *"activations are changing along a 1-d line"*, but is it not more accurate to say that the *variation* in the way activations are changing lies along a 1-d line? This brings me back to the bias term again. From my understanding, an effective rank of $1$ actually means that the activations tend to change in the direction given by $\textbf{b}$, while "spreading out" along some other direction given by the top principal component. This seems to be an important distinction to me, which surely changes the interpretation of most (if not all) of your results.

---

> > > ### Author Response · Authors · 2025-04-02
> > >
> > > Thank you for your reply. We couldn’t fully discuss this earlier due to the length limit. You gave a good example: after safety fine-tuning, activations shift toward a “harmlessness” directions on average. This global shift is captured by the bias term **b**. We agree with your example and we’d like to follow your example, explaining why we ignore **b**.
> > >
> > > To clarify, we decompose b into orthogonal directions: $b = \sum_{i=1}^n v_i$. We consider two types of $v_i$:
> > >
> > > - Type 1: $v_i$ is in W, meaning it has positive projections onto the safety residual space.
> > > - Type 2: $v_i$ is not in W, meaning it’s orthogonal to all feature directions and exists only in b.
> > >
> > > Since our methods focus on W, we have studied type 1 $v_i$ in our framework. Our main reason to ignore b is that **type 2 $v_i$ isn’t a feature direction, can’t be analyzed in our framework, and isn’t meaningful to study.**
> > >
> > > As type 2 $v_i$ isn’t in W, it’s not linearly correlate to input activations. Therefore it can’t be explained by linear input features. This contradicts the definition of a linear feature direction. As required by Definition 2.1, we cannot find a context pair that moves activations in this direction, as it’s independent of input context.
> > >
> > > Past researches has support that most meaningful feature directions **depend on specific inputs**. For example, [1] calculates the refusal direction by subtracting safe sample activations from unsafe samples. Similarly, [2] shows harmless output features are related to activation regions with toxic inputs, which DPO/SSFT learns to avoid.
> > >
> > > We understand and agree your example shows b can includes a global shift toward harmlessness. Past works[2] have supported your hypotheses. **But b representing a “harmlessness” direction doesn’t mean W fails to capture it**. As long as shifts in the “harmlessness” direction correlates with activations or input, which we believe holds for most safety-related directions, then W can incorporate it as a type 1 $v_i$. Our experiment in Figure 3 confirms this: the first component in W - I predicts model output with high accuracy.
> > >
> > > As an example, our training set contains both harmful and benign samples. A type 2 direction in b would means affecting the refusal rate regardless of content, which we think is less meaningful to study. Since it’s not triggered by input context, our PLRP method doesn’t apply, so we exclude it. **In summary, we ignores bias term b because it doesn’t fit in our framework, and we lose no meaningful directions by doing so**.
> > >
> > > On your **Q4** comment, thank you for the valuable question. As **b** is an activation offset, it shouldn’t affect the effective rank, which is a metric of variance. In practice, we compute the effective rank using W - I eigenvalues, also didn’t use **b**. We hope this clarification has make our answer to **Q4** clearer. We agree that the 1 rank might correspond to the direction of  **b**,  as normally direction of larger offset comes with larger variance. While this seems to support that key directions are captured by W, it requires further validation to confirm.
> > >
> > > We agree that measuring the global offset of activations (represented by b) is meaningful and provides important insights into the alignment mechanism. Previous studies have explored activation changes before and after safety fine-tuning (see our related work). In this paper, our contribution lies in applying PLRP and intervening on the decomposition of the safety residual space, making us the first to provide a detailed interpretation of ‘what is learned in the activation shift.’ In other words, we not only demonstrate that activations are pushed along safety-related directions but also offer a specific interpretation of those shifts.
> > >
> > > Thank you so much for your interest in our work! We hope above response has made the answer clearer. We’re more than happy to address any more questions or concerns you have.
> > >
> > > [1] Arditi, Andy et al. “Refusal in Language Models Is Mediated by a Single Direction.” ArXiv abs/2406.11717 (2024).
> > >
> > > [2] Lee, Andrew, et al. “A Mechanistic Understanding of Alignment Algorithms: A Case Study on DPO and Toxicity.” ICML 2024.

---

### Official Review · Reviewer_5f3T · 2025-03-12

**Overall Recommendation:** 3

**Summary:**

The paper focuses on the mechanisms of safety alignment in large language models, exploring how the internal representations of the model's refusal of harmful inputs manifest as multiple orthogonal directions in activation space. It introduces the concept of the "safety residual space" and identifies dominant and non-dominant directions through linear decomposition of representation shifts during the safety fine-tuning process. The paper designs and validates intervention experiments (such as removing specific trigger words) to reveal vulnerabilities in safety alignment, offering a new multidimensional perspective on understanding model safety.

**Claims And Evidence:**

I think claims are not entirely substantiated by the presented results and evidence.

The current analysis primarily focuses on a specific model (Llama 3 8B) and a specific dataset, and it remains unclear whether the findings can be generalized to other models or application scenarios.

**Essential References Not Discussed:**

No obvious omissions

**Experimental Designs Or Analyses:**

Yes

The current analysis primarily focuses on a specific model (Llama 3 8B) and a specific dataset, and it remains unclear whether the findings can be generalized to other models or application scenarios.

**Methods And Evaluation Criteria:**

Yes

**Other Comments Or Suggestions:**

Please read Question part.

**Other Strengths And Weaknesses:**

Strengths:
1. The paper proposes a multidimensional perspective on safety alignment and demonstrates through experiments the explanatory power and interactions of orthogonal directions in the safety residual space on model behavior.
2. The experimental results reveal that adjusting trigger words can bypass the model's safety defenses, providing new insights for improving model security.

Weaknesses:
1. The current analysis primarily focuses on a specific model (Llama 3 8B) and a specific dataset, and it remains unclear whether the findings can be generalized to other models or application scenarios.
2. Although the paper proposes methods for bypassing safety mechanisms, the discussion on how to improve model defenses based on these findings to prevent similar vulnerabilities is somewhat lacking.

**Questions For Authors:**

Are there any safety risks or side effects associated with intervention experiments on different non-dominant directions? How can intervention strategies be optimized to maintain both the overall capability and security of the model?

**Relation To Broader Scientific Literature:**

The paper introduces a multidimensional perspective on safety alignment, demonstrating through experiments how orthogonal directions in the safety residual space influence model behavior

**Theoretical Claims:**

I did not fully verify the correctness of the proofs for the theoretical claims.

---

> ### Author Rebuttal · Authors · 2025-03-31
>
> Thank you for your valuable suggestions. We will carefully address your questions in the following.
>
> > **Q1:** The current analysis primarily focuses on a specific model (Llama 3 8B) and a specific dataset, and it remains unclear whether the findings can be generalized to other models or application scenarios.
>
> Thank you for your insightful suggestions. We will answer your questions in two parts:
>
> **Generalization to other models.** Our evaluation focused on Llama 3.1 8B as a case study for mechanistic interpretation. We agree that evaluation on models of different scales would further strengthen our claims. To this end, we now include a safety residual space analysis for **Llama 3.2 3B** to verify that our findings generalize across scales. As shown in the figure below, the residual space for Llama 3.2 3B shows similar low-rank linearity and interpretability to Llama 3.1 8B.
>
> **[[View figure of analyzing Llama 3.2 3B]](https://cdn-uploads.huggingface.co/production/uploads/62cd3a3691d27e60db0698b0/3VaIR7muY-H_EalTIQhfh.png)**
>
> We will add this additional figure to the Appendix in the revised paper and reference it in the main body.
>
> **Generalization to other application scenarios.** Our current evaluation covers over 6 strong jailbreak baselines, representing different real-world adversarial attack strategies. By using jailbreak attack as an representative scenario, our works give mechanistic insight on safety features that can apply to other applications (e.g. backdoor attack or membership inference). We agree that expanding the scope of our safety residual space analysis to more scenarios would be valuable, and we plan to explore this in future work.
>
> > **Q2:** Although the paper proposes methods for bypassing safety mechanisms, the discussion on how to improve model defenses based on these findings to prevent similar vulnerabilities is somewhat lacking.
>
> Thank you for your insightful suggestions. In our conclusion, we briefly mentioned promising directions for improving model defenses based on the safety residual space. We can elaborate on two key methods:
>
> - **Interventions for Robustness.**  While interventions can remove safety ability by targeting specific directions in the residual space, they could also be used to strengthen defenses. Our framework identifies multiple crucial safety feature directions. This information could guide model editing techniques [1, 2] to selectively mitigate vulnerabilities by modifying features associated with specific non-dominant directions.
>
> - **LLM Red-Teaming.** Automated LLM Red-Teaming methods [3] could benefit from our findings too. Using PLRP on residual space components to identify trigger tokens offers a more mechanistic way to discover potential vulnerabilities. Red-teaming pipelines could incorporate our analysis to probe for weaknesses from internal representations more efficiently.
>
> We will incorporate a detailed discussion of potential defense improvements in the revision.
>
> > **Q3:** Are there any safety risks or side effects associated with intervention experiments on different non-dominant directions? How can intervention strategies be optimized to maintain both the overall capability and security of the model?
>
> Thank you for the valuable questions. We'd like to gently note that in our paper, intervention mainly serves as an **experimental tool** to validate the causal effect of non-dominant directions on safety behavior. We agree with you that there are potential risks and optimizations:
>
> - **Safety risks**: As demonstrated, targeted interventions removing specific non-dominant directions can weaken learned safety features, potentially offering a low-cost method to maliciously un-align a model.
> - **Side effects:** Appendix C.4 shows that interventions can slightly degrade general task performance, although in our experiments, the impact was relatively contained compared to alignment methods like DPO.
> - **Potential Optimization**: Our intervention generally follows established methods used for mechanistic analysis [4]. Optimizing inference-time interventions[5] is an active research area. Factors like intervention strength, targeted directions and the modules affected all affects the trade off. We are also interested in exploring this direction in future works.
>
> [1] Wei, Boyi, et al. "Assessing the brittleness of safety alignment via pruning and low-rank modifications." ICLR 2024.
>
> [2] Uppaal, Rheeya, et al. "Model editing as a robust and denoised variant of DPO: A case study on toxicity." ICLR 2025.
>
> [3] Lin, Lizhi, et al. "Against The Achilles' Heel: A Survey on Red Teaming for Generative Models." Journal of Artificial Intelligence Research 82 (2025).
>
> [4] Lee, Andrew, et al. "A mechanistic understanding of alignment algorithms: A case study on dpo and toxicity." ICML 2024.
>
> [5] Li, Kenneth, et al. "Inference-time intervention: Eliciting truthful answers from a language model." NIPS 2023.

---

> > ### Comment · Reviewer_5f3T · 2025-04-04
> >
> > Thanks for your explanations and additional experiments.
> >
> > I have no extra questions. I would be inclined to defer to the opinions of the other reviewers.

---

> > > ### Author Response · Authors · 2025-04-05
> > >
> > > Thank you for your valuable comments and suggestions! We will incorporate your suggestions to improve our paper.

---

### Official Review · Reviewer_wXRu · 2025-03-13

**Overall Recommendation:** 4

**Summary:**

This paper investigates which feature directions are used by safety-tuned language models to determine whether or not to refuse a request. This is done by optimizing an affine mapping to approximate the activations of a safety-tuned model given the corresponding activations in the pretrained model before safety tuning, referring to this map as the *safety residual space* (1). The paper presents a theorem stating that all directions in the safety residual space decrease the model's loss on net. The paper then safety-tunes LLaMA 3.1 models (one using DPO and one using self-supervised finetuning (SSFT)) and examine the safety residual spaces of these models. The paper looks at the "effective rank" of the safety residual space to quantify the extent of learned changes during safety-tuning at different layers, and finds differences between DPO and SSFT. It also looks at the singular vectors of the safety residual spaces, and uses a variant of layerwise relevance propagation to find patterns in which tokens are most important to these vectors, along with the importance of earlier-layer vectors to later-layer ones. The paper finds that ablating the first singular vector completely removes the model's refusal capability, while ablating later singular vectors cause the model to defend worse on specific jailbreak types. Finally, the paper examines jailbreak performance versues the number of jailbreak examples in the safety-tuning dataset, and finds that explicitly optimizing jailbreak examples based on information from the singular vectors makes it harder for the model to defend against them.

(1) Note that it's unclear to me whether the *safety residual space* refers to the map itself, to the matrix $W$ associated with the map (i.e. the linear part of the affine map), to the matrix $W-I$, or to the images of any of these maps. See "Other Strengths and Weaknesses" for more on this.

## Update after rebuttal

After the rebuttal process, I have changed my score to an accept. This is for the following reasons:

* The authors clarified all of the terminology in the paper that I found unclear, which was one of the biggest reasons for my initial low recommendation.
* The authors added new results regarding the accuracy of the learned linear transformation, filling one of the biggest gaps in their experiments.
* The authors removed the incorrect/superfluous theorem from the paper, and replaced it with an updated, more sensible framing.

The main idea of this paper -- that is, to investigate the difference between fine-tuned and base LLMs by learning an affine map between their activation spaces -- is very interesting, and now that the authors have addressed my concerns, I think that their paper putting forth this idea and performing investigations with it deserves to be accepted.

**Claims And Evidence:**

* In Section 4, the paper claims that "safety residual space is low-rank linear", on the basis of an empirically low effective rank of $W-I$, where $W$ is the matrix associated with the learned "safety residual space" affine map.
	* I believe that this claim is mostly supported. However, the paper never mentions the accuracy of the safety residual space map. If this map is inaccurate, then the results presented here might instead only mean that the *easily-linearly-approximable* changes in activations after safety-tuning are low rank -- while the "nonlinear" changes in activations could potentially be higher-dimensional. To address this, it would be useful for the paper to report the mean squared error between the activations of the safety-tuned model and the outputs of the safety residual space map, in addition to the mean squared norm of activations at this layer. (An even better (albeit slightly more complex) metric to compute would be the KL divergence between the safety-tuned model and the safety-tuned model whose activations at a given layer are replaced by the corresponding outputs from the safety residual space map.)
* In Section 5.1, the paper claims that certain directions are activated by specific patterns of tokens (e.g. "we observe that the dominant direction primarily relates to harmful subjects", "For instance, tokens like *Imagine*, *fictional* and *hypothetical* in L14-C2 establish a hypothetical tone"). However, from what I can tell, all of this is inferred by looking at relevance scores *on a single input example*, which is not enough to confidently make statements about the patterns that cause a direction to be activated. I recommend that the authors refer to "AutoInterp" methods (e.g. [1]).

All other claims in the paper seem well-supported as far as I could tell. The main other claims are as follows:
* Section 4: "The dominant direction predicts aligned behavior". The claim is supported by Figure 3, showing that the top principal components of the safety residual space achieve high classification accuracies of harmful vs. harmless behavior.
* Section 4: "multiple orthogonal feature directions can predict refusal behavior beyond the single dominant direction or probe vector". Mostly supported by the fact that as shown in Figure 3, early-layer "Best-of-N SSFT" predictions have accuracy around 80%, which is actually greater than the accuracy of the dominant direction predictor for these layers. (But the "Best-of-N SSFT" prediction accuracy no longer tells us anything about whether the non-dominant feature directions are useful predictors after around layer 15 or so, because the best-performing direction is the dominant direction. It would be better to exclude the dominant direction from among the Best-of-N SSFT directions.)
* Section 5.1: The paper claims that they found a specific direction L14-C6 which seems to activate on jailbreaks similar to those produced by PAIR. This claim is supported by evidence that ablating this direction increases jailbreak success rate on PAIR jailbreaks from ~0% to ~50%, without increasing other jailbreak success rates above ~10% (Figure 4).
* Section 5.2: "In most layers, directions primarily retain information from their counterparts in the previous layer. [...] In contrast, during early layers, directions exhibit a more dynamic pattern, receiving contributions from multiple directions in the previous layer.". This is well-supported by Figure 5, although explicit numerical metrics in the main text might further make this apparent.
* Section 6: The paper claims that running their layerwise relevance propagation variant on vectors in the "safety residual space" identifies tokens that induce refusal on the model. The paper supports this by automatically constructing jailbreaking prompts that avoid these tokens (using an algorithm dubbed "Trigger Removal"), and find that even when such prompts are included in a safety-tuning dataset, the model is still weaker to these jailbreaks than other jailbreaking methods when their outputs are included in the safety-tuning dataset (Table 2).

[1] https://transformer-circuits.pub/2023/monosemantic-features/index.html#global-analysis-interp-auto-acts

**Essential References Not Discussed:**

The main concept used throughout the paper's analysis is the "safety residual space", defined using a learned affine mapping from the activations of a base language model to the activations of a safety-tuned language model. While not directly applied to comparing safety-tuned models with their base model counterparts, there have been previous works that investigate the use of learned affine mappings between the activations of two different models, particularly [1] and [2]. The notion of the "safety residual space" can thus be viewed as an adaptation of this prior approach to the setting of understanding safety-tuned models, and I thus recommend that this paper cite these prior works.

[1] Lahner, Z., and Moeller, M. On the Direct Alignment of Latent Spaces. First Edition of the Workshop on Unifying Representations in Neural Models. 2023.

[2] Maiorca, V., Moschella, L., Norelli, A., Fumero, M., Locatello, F., and Rodola, E. Latent Space Translation via Semantic Alignment. 37th Conference on Neural Information Processing Systems. 2023.

**Experimental Designs Or Analyses:**

See "Claims and Evidence" and "Methods And Evaluation Criteria" for experiments whose validity I checked.

**Methods And Evaluation Criteria:**

**Weaknesses**

* In general, it is difficult for me to evaluate the datasets used by the paper for evaluation, since much of the paper is unclear about which datasets are even being used for evaluation (see Question #3 from later in this review).
* As someone unfamiliar with the layerwise relevance propagation literature, it was hard for me to evaluate the authors' extension of it, because the paper doesn't include a full description of their algorithm.
* I believe that the idea of the "safety residual space" presented in the paper is sound. However, no evaluation criteria (or actual evaluations) are provided for it. It is thus impossible to judge the accuracy of the learned affine map (see the "Claims and Evidence" section of this review).
* In Appendix C, the performance hit incurred by safety-tuned models is measured using perplexity increase, but this can be a difficult-to-interpret metric. Looking at model performance on datasets such as MATH, or alternatively using LLMs to evaluate outputs, might be better.
* In Section 4, it is never explained what the Best-of-N BASE baseline is. Additionally, it would be more sensible to exclude the dominant direction from the Best-of-N SSFT predictor, because once the dominant direction becomes the most effective predictor, the two have the same accuracies, preventing us from understanding the accuracy of the non-dominant directions.
* In Section 5, it is unclear how exactly partial layerwise relevance propagation scores are computed to me, a reader who is not familiar with the layerwise relevance propagation literature. I believe that readers who have a similar lack of familiarity with that area would be well-served by a detailed algorithm, possibly in the Appendix.
* In Section 5.1, the approach of interpreting directions based on a single input example seems ill-advised. I would recommend that for each direction, the authors compute the tokens across all input examples that have the highest PLRP scores; then, to find patterns and assess the interpretability of such patterns, an approach such as auto-interp [2] could be used.
* In Figure 6, the paper never says whether the plotted values are medians or means. Additionally, the figure gives no information about the distribution of the projection values (presumably dot products), even though it would be very useful to see whether these distributions have overlaps.

**Strengths**

* The safety-tuned models are trained via supervised finetuning and DPO. The resulting models were evaluated for ability to resist jailbreaks by measuring refusal percentage, and by using the "StrongREJECT evaluator" [1] to evaluate harmfulness of non-refused outputs. This is a standard, sensible evaluation.
* Using effective rank to measure the extent of the change in activations at different layers between the safety-tuned and base models (Fig. 2) is a very solid approach.
* In Section 5.2, the approach that the paper takes in determining the relevance of directions in adjacent layers is a sensible way of analyzing layerwise dependencies.
* The experiment on "safety finetuned models under different exposure settings" whose results are displayed in Table 2 is a sensible way to measure the importance of trigger tokens to jailbreaks.

[1] Souly, A., Lu, Q., Bowen, D., Trinh, T., Hsieh, E., Pandey, S., Abbeel, P., Svegliato, J., Emmons, S., Watkins, O., Toyer, S. A StrongREJECT for Empty Jailbreaks. arXiv preprint arXiv:2402.10260

[2] https://transformer-circuits.pub/2023/monosemantic-features/index.html#global-analysis-interp-auto-acts

**Other Comments Or Suggestions:**

I have none in particular.

**Other Strengths And Weaknesses:**

**Strengths**

The major strength of this paper, in my opinion, is the idea of learning an affine map between a pretrained and finetuned language model in order to mechanistically analyze the behaviors learned during finetuning. Even though learned maps between different models have already been considered in the literature (e.g. [1], [2]), the paper under review is the first that I've read that applies this to the same model pre- and post-finetuning. I could see this method being a useful tool for performing "model diffing"; it is thus to this paper's credit that it introduces such a method. I also like the idea of using a layerwise relevance propagation variant to assess which tokens are most important for causing each feature vector to activate. (Most mechanistic interpretability papers I've seen instead look at the tokens whose model activations have the largest dot products with the feature vector, which doesn't take into account tokens earlier in the context.)

**Weaknesses**

In addition to those addressed in earlier sections of this review (e.g. Theorem 3.2 seems to be incorrect, lack of information on evaluation datasets, etc.), one of the biggest is a pervasive lack of clarity and precision in the writing. Some examples of this are as follows:

* In lines 71-75, left column: "A one-dimensional *feature* value $W$ (e.g., "gender", "harmfulness") is defined as a latent variable that instantiates context $w$ through a generator function, denoted as $w := \mathcal{G}(W)$." It is not explained anywhere previously in the text what it means to instantiate context through a generator function, and I have not seen this verbiage used anywhere prior in the literature (including in the paper Park et al. (2023) cited in the sentence prior). My best guess is that the authors are trying to say that a feature is a latent variable that causally affects what output text $w$ is sampled from a model; sampling from the model under the effect of $W$ and obtaining $w$ as output is denoted by $w := \mathcal{G}(W)$.
* In lines 110-112, left column: "The safety residual space $V_\mathcal{T}$ is defined as the optimal affine transformation parameterized as $V_\mathcal{T} (x) = \mathbf{Wx} + \mathbf{b}$ that minimizes [...]". But in lines 128-129, the authors define $V_\mathcal{T}^{:k}$ to consist of the first $k$ singular components of $\mathbf{W} - \mathbf{I}$. Does this imply that $V_\mathcal{T}$ is equal to $\mathbf{W - I}$, is it equal to $\mathbf{W}$ (as the original definition suggested), or is it equal to the affine map $\mathbf{x} \mapsto \mathbf{Wx}+\mathbf{b}$?
* In the paragraph titled "Dominant Direction Predicts Aligned Behavior" in Section 4, it is stated that "components directly extracted from the trained models' activations fail to predict refusal behavior, as evidenced by the Best-of-N BASE baseline". However, it is never explained what the "Best-of-N BASE baseline" is, or what it means for components to be directly extracted from trained model activations. Given that they also use a predictor called "Best-of-N SSFT" which is described as using the "highest accuracy among the first 100 components for each layer", one might infer that "Best-of-N BASE baseline" refers to using the highest-accuracy component of among the first 100 components *in the base model*. But since the term "component" has thus far been used to refer to a singular vector of $W-I$ where $W$ is defined in terms of a map from base model to safety-tuned model activations, it is unclear what meaning this term has in the context of the base model alone.

These are a few examples of the overloaded/inconsistent terminology present all throughout the paper. (Due to the quantity of such instances, I cannot list or go into detail on every single example; note that some more are mentioned in the "Questions" section below.) Because this sort of writing is consistently present throughout the paper, it was thus very difficult for me to parse what methods exactly were being used or what exactly was being studied in certain experiments.

I recommend that the authors reread this paper carefully, making sure that whenever a technical term is used (e.g. "residual space", "component"), it is true that 1) the term is defined somewhere and that 2) the current use of the term is consistent with the definition and with previous use. If the authors do this and revise the paper accordingly, I am confident that readers would be far better able to appreciate the good ideas presented in this paper.

[1] Lahner, Z., and Moeller, M. On the Direct Alignment of Latent Spaces. First Edition of the Workshop on Unifying Representations in Neural Models. 2023.

[2] Maiorca, V., Moschella, L., Norelli, A., Fumero, M., Locatello, F., and Rodola, E. Latent Space Translation via Semantic Alignment. 37th Conference on Neural Information Processing Systems. 2023.

**Questions For Authors:**

Note that the following list of questions is not exhaustive. Nevertheless, I believe that the paper should be edited with answers to them before I would consider raising my score.

1. Does the term "safety residual space" refer to the affine map from base model activations to safety-tuned model activations given by $x \mapsto Wx + b$, does it refer to only the linear part of the map $W$, or does it refer to $W-I$?
2. How was the probe vector in Section 4 computed? Was it computed via the same method on the same dataset as Arditi et al. 2024, or was anything different in the authors' setup in this paper?
3. In general, I couldn't find a clear statement of what dataset (and how many examples, etc.) was used for learning the safety residual space map, for evaluating effective rank (Fig. 2), and for measuring prediction accuracy (Fig. 3). Is this the same dataset that was used for finetuning the models in Section 3.2?
4. In Section 4, what exactly is the Best-of-N BASE baseline and how is it computed?
5. In Section 5, when PLRP is used, is the relevance value associated with each context token equal to the relevance of that context token for causing the direction to have a high dot product with a fixed token's activations? Is the relevance value with respect to the direction's dot product with all tokens in the context? Or something else? (More generally, I am personally not familiar with LRP, so I would find it useful for myself and for other readers unfamiliar with that specific approach to add a full description of the PLRP formulas somewhere in the paper or in an appendix.)
6. In Section 6, when describing dominant component suppression, the paper states "To preserve the model’s ability to refuse plainly harmful prompts, we exclude component directions with harmfulness correlations above 0.7." Does the term "harmfulness correlation" mean "correlation between the dot products of the vector with model activations and the binary variable of whether the input is harmful"?
7. In Section 6, how many non-dominant components are suppressed? Are all non-dominant components with harmfulness correlations above 0.7 suppressed, or only some?

The following question is less vital to be answered, but I would still be interested in knowing the answer:

1. In Section 4, it is stated that "all components found have near-zero cosine similarity with the probe vector". What about if the mean of these components is taken -- does the mean have a higher cosine similarity? Less importantly, what are the exact cosine similarities? (It would make the most sense to put these values in an appendix rather than the main body.)

**Relation To Broader Scientific Literature:**

This paper is related to understanding at a mechanistic level the behavior induced by safety-tuning. It is related to papers that directly focus on understanding the activation spaces of safety-tuned models, such as [1] (which this paper cited), that finds a direction in the activation spaces of safety-tuned models that mediates whether harmful requests are refused or not. It is also related to papers that study this question from the perspective of parameter space, such as [2], which finds that safety-tuned models' parameters live in small "basins" in parameter space outside of which model harmfulness increases drastically.

There is also some relationship between the methods and findings introduced in this paper and some previous literature in model comparison, which will be further elaborated upon in the "Essential References Not Discussed" section in this paper.

[1] Arditi, A., Obeso, O., Syed, A., Paleka, D., Panickssery, N., Gurnee, W., and Nanda, N. Refusal in language models is mediated by a single direction. arXiv preprint arXiv:2406.11717, 2024.

[2] Peng, S., Chen, P., Hull, M., Chau, and Chau, D. Navigating the Safety Landscape: Measuring Risks in Finetuning Large Language Models. 38th Conference on Neural Information Processing Systems, 2024.

**Theoretical Claims:**

I believe that the proof of Theorem 3.2 as given in Appendix A is incorrect. The offending lines are the following:

> Given that $W^*$ converged with $J(W^*) \ge \sup_{W \in \mathcal{M}} J(W) - \epsilon$, it cannot contain any zero-utility directions. If it did, we could remove such a direction without affecting performance, contradicting the convergence assmuption.

This is false: convergence to a local optimum does not imply that any change in the weights would yield a worse-performing network. A near-trivial example of this (for ReLU networks) is given in [1]: "in ReLU networks we can multiply the weights in one layer by a certain value, and divide the weights in the next layer by the same value, and the outputs of the network (and consequently loss) wouldn’t change." (In the associated paper [2], the authors of [1] find more complex examples of this phenomenon, known as "mode connectivity" in the literature and believed to be an important factor in why neural networks are able to generalize.)

In fact, even if we did choose to make the stronger assumption (which is not empirically borne out in real neural networks) that there is no mode connectivity between the optimal solution $W^*$ and any other solutions, then the argument made in the proof would imply that *any* nonzero vector $v$ (rather than solely directions in the safety residual space) must contribute positively to the training objective on some input (in the sense that ablating $v$ from the model's weights would cause the model to perform worse on that input). The resulting conclusion would thus be trivial.

[1] https://izmailovpavel.github.io/curves_blogpost/

[2] Garipov, T., Izmailov, P., Podoprikhin, D., Vetrov, D., and Wilson, A.
Loss Surfaces, Mode Connectivity, and Fast Ensembling of DNNs. Advances in Neural Information Processing Systems 31. 2018.

---

> ### Author Rebuttal · Authors · 2025-03-31
>
> We thank the reviewer for the thorough and insightful feedback. We will clarify all your questions in the revised paper. We will start by answering your questions and then address the remaining issues.
>
> > **Q1:** Does the term "safety residual space" refer to the affine map, linear $W$ or $W-I$?
>
> According to Definition 3.1, **the safety residual space $V_T$ was defined as an affine map**. Separately, we used $V^{:k}_T$ to represent the first k right singular vectors of $W - I$. We recognize that overloading $V_T$ caused confusion. Following your feedback, we will:
>
> -   Define the safety residual space map with a distinct symbol, changing $V_T(X)$ to $S(X)$.
> -   Consistently use $V$ only for the singular vectors of $W-I$.
>
> > **Q2:** How was the probe vector in Section 4 computed?
>
> We applied the exact method described in Arditi et al. (2024) to our training dataset. While the resulting probe vector has different values than in Arditi et al. (2024) due to the different dataset, it carries the same meaning.
>
> > **Q3:** I couldn't find a clear statement of what dataset was used for learning the safety residual space map and evaluation.
>
> Thank you for pointing this out. As stated in L124 left, **the *same* training dataset (2600 samples, detailed in Figure 7) was used for both finetuning the model and learning the map**: *"We compute activations from the training data as an approximation of the unaligned distribution $X_u$"*. Specifically, this training set was used to learn the residual space map (Figure 2). All prediction and intervention evaluations (Figures 3-6) were performed on the test set. We will state this more prominently.
>
> > **Q4:** In Section 4, what exactly is the Best-of-N BASE baseline and how is it computed?
>
> Apologies for the lack of clarity. As referenced in L195 right ("components directly extracted from the trained models’ activations..."), we follow the process described in L126 left, using SVD to extract components. **We perform SVD on the stacked base model activations from the training data and select the singular vector yielding the highest refusal prediction accuracy on the test set**. We will add this explicit definition.
>
> > **Q5:** What does relevance compute in PLRP?
>
> PLRP computes the relevance of each context token for a direction vector (e.g., L14-C1) having a high dot product onto the activation at the *last token position* in the specified layer (e.g., Layer 14). The relevances are back-propagated from this single position to all context tokens. We described PLRP in Section 5, but acknowledging that LRP methods might be unfamiliar, we will add a more detailed algorithm description and background in the Appendix, as suggested.
>
> > **Q6:** Does the term "harmfulness correlation" mean ...
>
> Yes, that is correct. We will revise the text to use this clearer phrasing.
>
> > **Q7:** Section 6, how many non-dominant components are suppressed?
>
> This is a good question. We will add the following figure in the revision to clarify. Please refer to the caption.
>
> **[[View figure of harmfulness correlation]](https://shorturl.at/ncMfk)**
>
>
> **Other Issues**
>
> > 1. No evaluation criteria are provided for accuracy of the safety residual space map
>
> You raise a good point. We will report these evaluation results in the revision. Due to length limit, please see our rebuttal to Reviewer 6Hnk **Q1**.
>
> > 2. The activation pattern is inferred by looking at relevance scores _on a single input example.
>
> We will add an aggregated evaluation to strengthen our claims. Please see rebuttal to Reviewer 6Hnk **Q6**.
>
> > 3. In Figure 6, the paper never says whether the plotted values are medians or means and ...
>
> The plotted values are **means** across the test dataset. We will clarify this in the caption and add the following figure showing distributions to the Appendix.
>
> **[[View figure of projection distribution in Figure 6]](https://shorturl.at/4HzSc)**
>
> > 4. The proof of Theorem 3.2 as given in Appendix A is incorrect
>
> Your analysis is insightful. We agree that the convergence assumption does not preclude zero-utility directions, especially considering mode connectivity. **We will correct the theorem's statement and proof**. Specifically, we will change the quantification from "For any residual direction $v \in V_T$ ..." to "There exist residual directions $v \in V_T$..." and update the proof accordingly.
>
> > 5. Essential References Not Discussed
>
> Thank you for highlighting these references. We will incorporate citations and discussion into the Related Work section, and we will **definitely** perform a careful pass to ensure all technical terms are clearly defined upfront and used consistently.
>
> Nevertheless, we are grateful the reviewer found our idea novel and supported by solid experiments. We hope these clarifications and proposed revisions address your concerns, and we would appreciate it if you would reconsider your evaluation based on our response.

---

> > ### Comment · Reviewer_wXRu · 2025-04-04
> >
> > Thank you for taking the time to address my concerns. The clarifications that you provided in your rebuttal definitely help my understanding of the paper (and hopefully will help other readers as well), and the additional results very much strengthen your argument. (The high interpretability of the top aggregated LRP tokens in particular was very exciting to see.)
> >
> > My main remaining sticking point with the paper is Theorem 3.2. I believe that the paper would be improved by omitting it entirely, since in its corrected form, the result is essentially trivial. If anything, it only distracts from the interesting empirical results that you have found.
> >
> > Given your rebuttal, I am changing my recommendation from Reject to Weak Accept. Removing Theorem 3.2 would cause me to change my recommendation to Accept.

---

> > > ### Author Response · Authors · 2025-04-05
> > >
> > > We sincerely thank you again for the thoughtful and constructive suggestions, which have significantly contributed to the paper. We understand and agree with your suggestions that the updated theorem 3.2 has limited practical implications for the experiment and results. In this paper, our theorem mainly provided insight and motivation for studying components in the residual space as feature directions, bridging the two concepts. **Following your suggestion, we will revise and replace Theorem 3.2 with a more direct and qualitative hypothesis**. Specifically, we will update Section 3.1 as follows: (Note $V_\mathcal{T}$ follows the revised definition, singular vectors of $W-I$)
> > >
> > > >A key question is whether the components in the residual space contain interpretable features, similar to probe vectors. Conceptually, the safety finetuning optimizes the model to produce safer outputs. This process induces activations to shift along specific directions to align with safety objectives, which we capture with $V_\mathcal{T}$. These direction in $V_\mathcal{T}$ are strong candidates for feature directions under the definition in Equation 1, as they increase probability of safe output when activations are moved along those directions. While this does not guarantee human-interpretable features, it suggests $V_\mathcal{T}$ is a promising source for automatically discovering safety-related feature directions without requiring probing data pairs. To generalize this idea, we have the following hypothesis:
> > > >
> > > >
> > > >**Hypothesis 1. (Finetuning Residuals as Feature Directions)** The principal components representing the activation shifts induced by safety finetuning contains safety-related feature directions. Furthermore, orthogonal directions within this space potentially represent distinct and interpretable safety features.
> > > >
> > > >
> > > >In the following sections, we verify this hypothesis by examining the top components of $V_\mathcal{T}$. We study (1) if the components in $V_\mathcal{T}$ are feature directions and (2) what specific features these directions represent.
> > >
> > > We will also update the introduction and remove the proof in the appendix accordingly. In this way, **we can preserve the theoretical insight while focusing more on the empirical evidence**, which we believe will better highlighting our findings and contributions.

---

### Official Review · Reviewer_3Lkg · 2025-03-14

**Overall Recommendation:** 3

**Summary:**

This paper investigates the multi-dimensional nature of safety-aligned behaviors in LLMs, challenging the traditional single-direction representation of safety features. The authors introduce the concept of a safety residual space, analyzing activation shifts during safety fine-tuning of Llama 3 8B. Through singular vector decomposition and layer-wise relevance propagation, they identify a dominant refusal direction alongside smaller, interpretable directions related to role-playing and hypothetical narratives. The study further explores how these secondary directions influence refusal behavior and demonstrates that removing trigger tokens from harmful prompts can weaken safety alignment, highlighting potential vulnerabilities.

**Claims And Evidence:**

Overall, the paper presents strong and convincing evidence for its claims. The findings are well-supported by both theoretical derivations and empirical experiments, though some assumptions (e.g., the full linearity of safety features) may require further scrutiny. Expanding validation beyond Llama 3 and incorporating potential countermeasures against jailbreak attacks would further enhance the robustness of the study.

**Essential References Not Discussed:**

None

**Experimental Designs Or Analyses:**

The evaluation of Vt is well-structured, systematically examining whether all retained directions contribute positively to the training objective. However, it would be beneficial to include an ablation study that explicitly removes certain directions and observes the impact on model performance, further reinforcing the claim that all retained directions are necessary. Regarding dataset construction, the inclusion of multiple jailbreak methods ensures a diverse set of adversarial examples, making the evaluation more comprehensive. However, further details on data balancing and sampling strategies could be provided to ensure fairness in evaluating different attack methods. Overall, the experimental design is well-motivated and provides strong empirical support for the theoretical claims. Small refinements, such as broader model comparisons, could further enhance the robustness of the conclusions.

**Methods And Evaluation Criteria:**

The chosen methods and evaluation criteria are appropriate for the research problem, providing valuable insights into multi-dimensional safety representation in LLMs. However, further validation on larger models and real-world safety scenarios would strengthen the generalizability of the findings.

**Other Comments Or Suggestions:**

refer to the weakness

**Other Strengths And Weaknesses:**

Strengths:
1.	Originality: The integration of safety fine-tuning and residual shift analysis to address jailbreak attacks is a novel contribution. The method of iterative trigger removal is innovative and adds value to the field.
2.	Practical Relevance: The method directly addresses the growing concern of adversarial robustness in large language models, making it highly relevant for both academic and industry applications.
Weaknesses:
1.	Limited Evaluation: The empirical evaluation is not extensive enough. More detailed experiments and comparisons with existing methods are needed to strengthen the paper's claims.
2.	Scalability: The iterative trigger removal process may be computationally expensive. A discussion of scalability and potential optimizations would improve the paper.
3.	Notation Clarity: Some mathematical terms and notations, such as zero-utility directions and residual shifts, could be clarified for better accessibility.

**Questions For Authors:**

refer to the weakness

**Relation To Broader Scientific Literature:**

This paper makes a meaningful contribution by extending prior work in LLM security and robustness while introducing novel insights into the structure of residual shifts and their role in adversarial mitigation.

**Theoretical Claims:**

The paper provides a formal proof of Theorem 3.2, establishing that a converged model does not retain zero-utility directions in the learned residual shift space. The theoretical claims appear logically valid and are backed by rigorous derivations.

---

> ### Author Rebuttal · Authors · 2025-03-31
>
> Thank you for the constructive feedback. We appreciate that the reviewer found our core claims well-supported and our analysis novel. We will address each of your concerns below.
>
> >**Q1:** Limited Evaluation: The empirical evaluation is not extensive enough.
>
> Thank you for these valuable suggestions. We will address each point individually.
>
> **Evaluation on larger models (Methods And Evaluation Criteria).** Our evaluation focused on Llama 3.1 8B as a case study for mechanistic interpretation. We agree that evaluation on models of different scales would further strengthen our claims. To this end, we now include a safety residual space analysis for **Llama 3.2 3B** to verify that our findings generalize across scales. As shown in the figure below, the residual space for Llama 3.2 3B shows similar low-rank linearity and interpretability to Llama 3.1 8B, which clearly support the generalizability of our findings.
>
> **[[View figure of analyzing Llama 3.2 3B]](https://cdn-uploads.huggingface.co/production/uploads/62cd3a3691d27e60db0698b0/3VaIR7muY-H_EalTIQhfh.png)**
>
> We will add this additional figure to the Appendix in the revised paper and reference it in the main body.
>
> **Evaluation on real-world safety scenarios and comparisons with existing methods. (Methods And Evaluation Criteria)** Thank you for this insightful feedback. Our current evaluation covers over 6 strong jailbreak baselines, representing different real-world adversarial attack strategies. We also compared the effectiveness of our trigger removal attack against these jailbreak methods. We agree that expanding the scope of our safety residual space analysis to more scenarios would be valuable, and we plan to explore this in future work.
>
> **Evaluation on the full linearity of safety features. (Claims And Evidence)** Thank you for this insightful suggestion. Our framework does not strictly require that safety features are fully linear. In the following table, we evaluate how well activation shifts during safety finetuning can be approximated by linear affine maps. We found that the Mean Square Error between finetuned activations and the approximation is negligible compared to the Norm of activations. This suggests **safety finetuning mostly introduces linear safety features**.
>
> **[[View table of residual space map MSE]](https://cdn-uploads.huggingface.co/production/uploads/62cd3a3691d27e60db0698b0/CPfaKqhmqgHIv9V9HRx8j.png)**
>
> We will report these results for our learned linear map in the revision to strengthen the claims regarding the linearity of safety features.
>
> > **Q2:** Scalability: The iterative trigger removal process may be computationally expensive.
>
> Thank you for this suggestion. While iterative trigger removal serves mainly as an analytical tool, **its computational cost (at most 30 attempts per sample in our settings) is comparable to other iterative methods** like TAP [1] (avg. 35 attempts) and PAIR [2] (avg. 37 attempts). The computational requirement is feasible in practice. We will add a comparison to these baselines in the paper, following the description of the trigger removal algorithm.
>
> > **Q3:** It would be beneficial to include an ablation study that explicitly removes certain directions.
>
> Thank you for pointing this out. We believe there may be a misunderstanding, as we did provide an ablation study removing certain directions to measure changes in model behavior (Figure 4 for refusal ability and Appendix C.4 for general ability). We will make the reference to this study clearer in the revision.
>
> > **Q4:** Further details on data balancing and sampling strategies could be provided.
>
> Thank you for your constructive suggestions. Details on our data balancing and sampling processes are provided in Appendix C.1. We will add a more direct reference to this section in the main text to make this information more visible.
>
> > **Q5:** Notation Clarity: Some mathematical terms and notations, such as zero-utility directions and residual shifts, could be clarified for better accessibility.
>
> Thank you for pointing this out. We will clarify these terms in the revised paper to improve accessibility:
>
> -   *Residual shifts* refer to the change in activations within the safety residual space before and after finetuning.
> -   *Zero-utility directions* refer to directions in the safety residual space that do not contribute to changing the model's safety behaviors.
>
> In the revision, we will add these definitions where the terms are first introduced.
>
> [1] Mehrotra, Anay, et al. "Tree of attacks: Jailbreaking black-box llms automatically." Advances in Neural Information Processing Systems 37 (2024): 61065-61105.
>
> [2] Chao, Patrick, et al. "Jailbreaking black box large language models in twenty queries." arXiv preprint arXiv:2310.08419 (2023).

---

### Decision · Program_Chairs · 2025-05-01

**Decision:**

Accept (poster)

**Comment:**

This paper analyzes the changes in the activation space after safety fine-tuning. They define the framework of Safety Residual Space and using this framework, they show that a dominant direction governs the model’s refusal behavior. They also show that by removing some directions, they model's safety behavior improves.

This is a solid contribution to the field of mechanistic interpretability. I encourage the authors to address reviewers' remaining concerns in the camera ready version.